# Systematic evaluation of multifactorial causal associations for Alzheimer's disease and an interactive platform MRAD developed based on Mendelian randomization analysis

**Tianyu Zhao[1], Hui Li[2,3], Meishuang Zhang[4], Yang Xu[1], Ming Zhang[1], Li Chen[1]\***

[1]Department of Pharmacology, College of Basic Medical Sciences, Jilin University, Changchun, China; [2]Department of Neurology, Xuanwu Hospital, Capital Medical University, Beijing, China; [3]Neurology and Intracranial Hypertension & Cerebral Venous Disease Center National Health Commission of China, Xuanwu Hospital, Capital Medical University, Beijing, China; [4]School of Nursing, Jilin University, Changchun, China

**\*For correspondence:**
chenl@jlu.edu.cn

**Competing interest:** The authors declare that no competing interests exist.

**Abstract** Alzheimer's disease (AD) is a complex degenerative disease of the central nervous system, and elucidating its pathogenesis remains challenging. In this study, we used the inverse-variance weighted (IVW) model as the major analysis method to perform hypothesis-free Mendelian randomization (MR) analysis on the data from MRC IEU OpenGWAS (18,097 exposure traits and 16 AD outcome traits), and conducted sensitivity analysis with six models, to assess the robustness of the IVW results, to identify various classes of risk or protective factors for AD, early-onset AD, and late-onset AD. We generated 400,274 data entries in total, among which the major analysis method of the IVW model consists of 73,129 records with 4840 exposure traits, which fall into 10 categories: Disease, Medical laboratory science, Imaging, Anthropometric, Treatment, Molecular trait, Gut microbiota, Past history, Family history, and Lifestyle trait. More importantly, a freely accessed online platform called MRAD (https://gwasmrad.com/mrad/) has been developed using the Shiny package with MR analysis results. Additionally, novel potential AD therapeutic targets (CD33, TBCA, VPS29, GNAI3, PSME1) are identified, among which CD33 was positively associated with the main outcome traits of AD, as well as with both EOAD and LOAD. TBCA and VPS29 were negatively associated with the main outcome traits of AD, as well as with both EOAD and LOAD. GNAI3 and PSME1 were negatively associated with the main outcome traits of AD, as well as with LOAD, but had no significant causal association with EOAD. The findings of our research advance our understanding of the etiology of AD.

## eLife assessment

This **important** study introduces the MRAD database, an advancement in Alzheimer's disease research that provides a powerful tool for evaluating risk and protective factors through Mendelian randomization analysis. The evidence supporting the database's utility is **solid**, with findings backed by robust data, though addressing methodological concerns and ensuring more rigorous validation of associations would further strengthen its impact. This resource represents a significant leap forward in the field, offering unprecedented opportunities for researchers and clinicians to uncover key insights into Alzheimer's etiology, potentially revolutionizing how

Alzheimer's research is approached and accelerating the discovery of new prevention strategies and treatments.

## Introduction

AD is a progressive degenerative disease of the central nervous system, characterized by cognitive impairment, reduced functional capacity for daily living, and behavioral changes. It can be divided into two types: early-onset AD (EOAD, age of onset ≤65 years) and late-onset AD (LOAD, age of onset >65 years); the proportion of LOAD in patients with AD is approximately 95%, with LOAD having a stronger genetic predisposition than EOAD (*Godyń et al., 2016*; *Ransohoff and El Khoury, 2015*; *Zou et al., 2014*). According to the latest data from the World Health Organization (WHO), the population with AD is currently over 50 million worldwide and is expected to rise to 115 million by 2050 (*GBD and Dementia Forecasting Collaborators, 2019*; *Anonymous, 2022*). With the increasing aging population, the incidence of AD continues to rise, making AD the fifth leading cause of death worldwide. Given that AD is a chronic complex disorder involving multiple pathophysiological changes, it is likely caused by the joint action of various factors in a multifaceted pathological process, and this intricate nature of AD contributes to the current challenges in its diagnosis and treatment, such as low consultation rates, high rates of misdiagnosis at initial consultations, and low rates of long-term standardized treatment (*World Alzheimer Report, 2022*), thereby making AD one of the most perplexing diseases. Consequently, examining the pathogenic mechanisms of AD, identifying its risk factors, and conducting timely and effective early screening and diagnosis are of utmost importance.

Traditional epidemiological studies have reported common risk factors for AD. Some metabolic co-morbidities are highly associated with AD, such as cardiovascular disease (*Falsetti et al., 2018*; *Femminella et al., 2018*), obesity (*Pegueroles et al., 2018*; *Anstey et al., 2013*), and diabetes (*Jayaraman and Pike, 2014*; *Vagelatos and Eslick, 2013*). Serological parameters such as C-reactive protein (*Cooper et al., 2023*), lipids (*Xu et al., 2020*; *Zhu et al., 2022*), and vitamin levels *Lopes da Silva et al., 2014*; *Yu et al., 2020*; *Douaud et al., 2013* have been previously reported as potential biomarkers for AD. In addition, some factors related to lifestyle, family history, education, economic level, and environment correlate with AD (*Kivipelto et al., 2018*; *Wei et al., 2015*; *Livingston et al., 2017*; *Livingston et al., 2020*). However, most epidemiological studies are insufficient to draw definitive conclusions on causal association due to the potential for reverse causality and confounding bias.

MR analysis is an emerging method to explore the causal association between AD and various factors (*Hu et al., 2022*; *Williams et al., 2020*; *Cui et al., 2022*). MR analysis reduces confounding and reverse causality due to the segregation and independent assortment of genes passed from parents to offspring (*Hingorani and Humphries, 2005*). In the absence of pleiotropy (that is, genetic variation related to a disease via other pathways) and demographic stratification, MR can present a clear estimate of risk of the disease (*Emdin et al., 2017*; *Davey Smith and Hemani, 2014*). MR analysis is increasingly used to determine a causal relationship between potentially modifiable risk factors and outcomes (*Davies et al., 2018*). These advantages make MR a valuable tool to better elucidate the potential risk or protective factors for AD.

*Chen et al., 2023* used MR analysis to reveal the causal relationship between AD and factors including sociodemographic and early life status. However, the study revealed they are restricted by the available variables from the UKB database, which leads to variables such as air pollution, blood glucose measures, and so on were not included. And also, due to the high degree of heterogeneity present in AD subtypes, EOAD and LOAD have different biological and genetic characteristics. Our study uses the MRC IEU OpenGWAS database as the sample source for MR analysis to address the aforementioned limitations. The MRC IEU OpenGWAS database, the largest open GWAS database globally, has compiled 42,335 GWAS summary datasets from sources such as the UK Biobank, FinnGen Biobank, and Biobank Japan. Analyzing large-scale datasets will break new ground for MR research on AD.

MR requires a combination of background knowledge in biology, computer science, software studies, and statistics, which often leads to a dilemma where biologists are not well-versed in computer and statistical fields, while computer science experts struggle to adopt a medical biology mindset. Consequently, the vast majority of available GWAS data have not been effectively utilized through MR. Therefore, the construction of a multi-level data platform specifically for AD based on

MR analysis of massive GWAS data is of great strategic significance, and it will facilitate researchers and clinicians worldwide to conveniently and rapidly obtain risk factors that are causally associated with AD.

In summary, in this work we attempt to identify risk or protective factors causally associated with AD from a holistic and systematic perspective, thereby providing new ideas for understanding the AD pathogenesis, achieving early diagnosis, and developing clinical drugs. In the first place, this study uses a hypothesis-free data mining approach to studying the possible etiology of Alzheimer's disease based on MR, with specific attention to different AD subtypes (EOAD and LOAD). Based on this, we developed an online open integrated platform, MRAD (Mendelian randomization for Alzheimer's disease, https://gwasmrad.com/mrad/). Moreover, the platform was further enriched by including related targets' information such as functions and pathways retrieved from the public database Uniprot. The platform is the first multi-dimensional, integrated, shared, and interactive comprehensive platform for AD MR research to date.

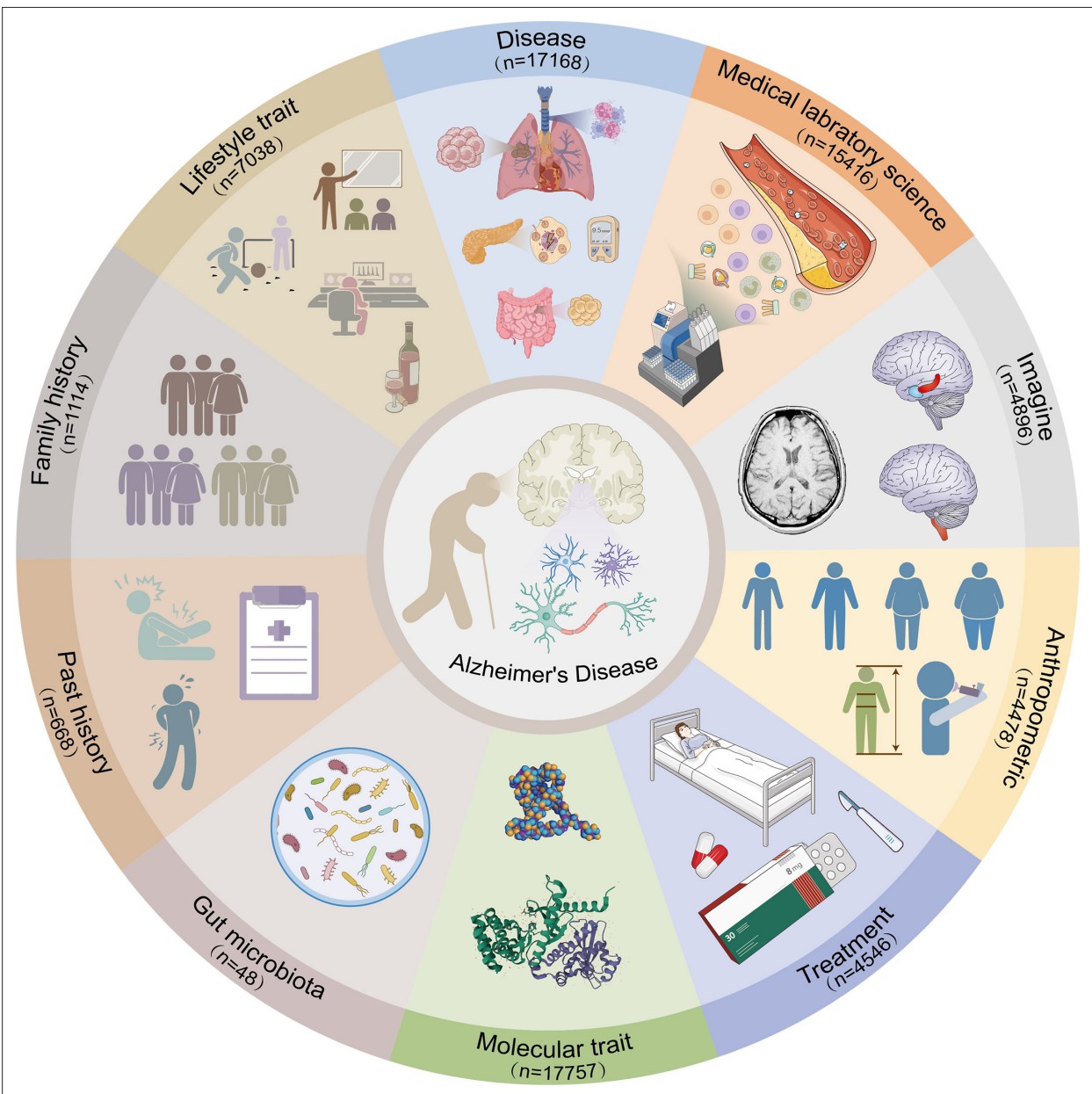

**Figure 1.** Categories of the exposure traits identified by the inverse-variance weighted (IVW) model.

## Results

### Results of hypothesis-free Mendelian randomization analysis for Alzheimer's disease

Based on hypothesis-free Mendelian randomization analysis for Alzheimer's disease, this study generated a total of 400,274 data points. The major analysis method of the IVW model consists of 73,129 records with 4840 exposure traits, which fall into 10 categories: Disease (n=17,168), Medical laboratory science (n=15,416), Imaging (n=4,896), Anthropometric (n=4478), Treatment (n=4546), Molecular trait (n=17,757), Gut microbiota (n=48), Past history (n=668), Family history (n=1114), and Lifestyle trait (n=7038), as shown in *Figure 1*. To assess the robustness of the IVW results, sensitivity analysis was performed using six other models (MR-Egger with a total of 50,804 records, Weighted median with a total of 50,804 records, Simple mode with a total of 50,804 records, Weighted mode with a total of 50,804 records, Maximum likelihood with a total of 73,125 records, and Penalized weighted median with a total of 50,804 records).

### MRAD platform integration

Based on the 400,274 data points stated above, we created herein is an online data analysis platform for identifying the risk or protective factors for AD called MRAD (Mendelian randomization for Alzheimer's disease, https://gwasmrad.com/mrad/). It contains six modules: (i) Home; (ii) Study Design; (iii) IVW interactive; (iv) IVW static; (v) Sensitivity analysis interactive; and (vi) Sensitivity analysis static; The platform provides a user-friendly search interface, allowing users to search, interactively visualize, analyze, and download the obtained results. MRAD User Guide is provided below in detail.

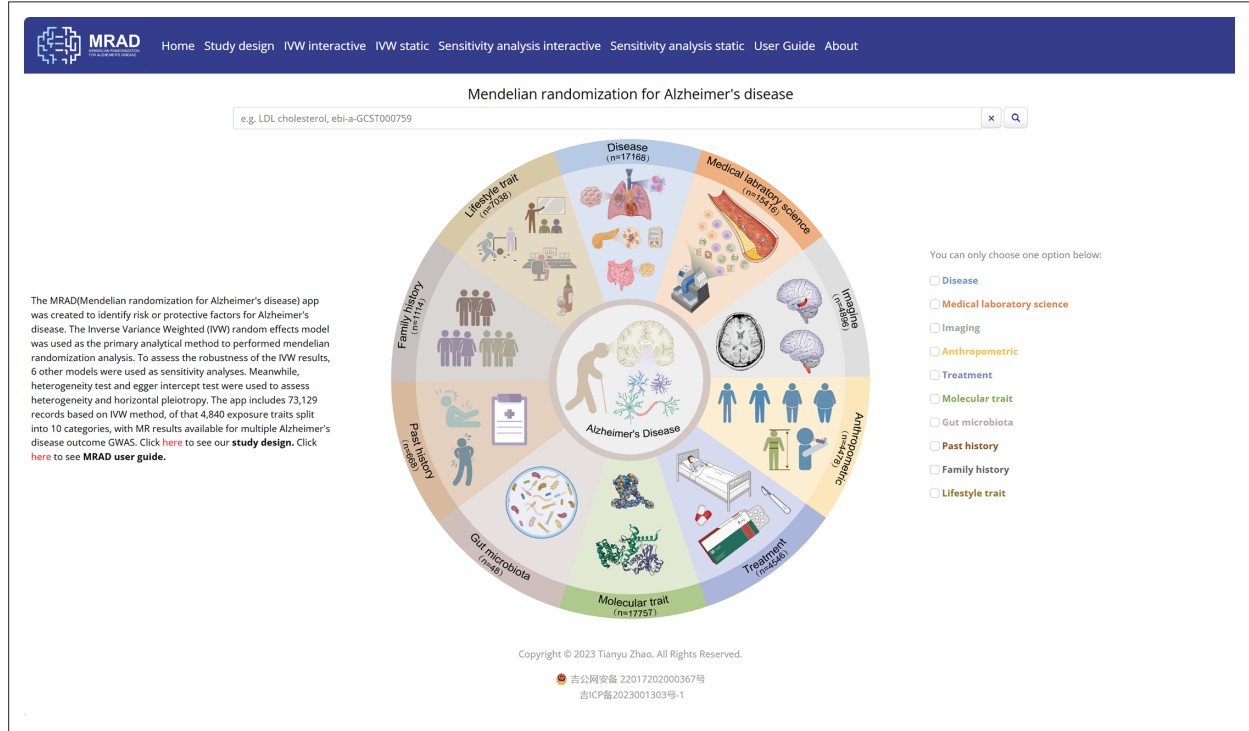

**Figure 2.** Home module.

The online version of this article includes the following source data and figure supplement(s) for figure 2:

**Source data 1.** Inverse-variance weighted (IVW) interactive module.

**Source data 2.** Sensitivity analysis interactive module.

**Figure supplement 1.** 400,274 records search interface of Mendelian randomization for Alzheimer's disease (MRAD).

**Figure supplement 2.** Study design module.

**Figure supplement 3.** Interactive visual results in the inverse-variance weighted (IVW) interactive module.

**Figure supplement 4.** Interactive visual results in the sensitivity analysis interactive module.

## Home module (as shown in Figure 2)

The Home module allows users to search through 400,274 records. Users can enter search terms, such as exposure (e.g. LDL cholesterol) or id.exposure (e.g. ebi-a-GCST000759), in the search box. The search interface includes 20 control widgets, as shown in *Figure 2—figure supplement 1*. Among these control widgets, there are eight select boxes, including Classification I, Classification II, Classification III, id.outcome, method, effect_direction, Consortium, and Sex, allowing users to make single or multiple selections to filter the data. The control widgets also include two text input boxes, id.exposure and exposure, which allow users to search by text input (case-insensitive). The control widgets Min pval and Max pval allow users to search by numerical input (initial values: Min pval = 0, Max pval = 1). The Heterogeneity test and MR-Egger intercept test checkboxes provide users with a single-click option to obtain data results that have passed heterogeneity and horizontal pleiotropy tests (p>0.05). The Year range slider helps users to filter data based on the year of publication (range: 2007–2022). Additionally, we have supplemented relevant basic information on biomarker functions and pathways from the public database Uniprot. Users can search for information using the Uniprot Entry ID and Gene Names input boxes and can click on the Uniprot Entry ID Link in the table to go to the corresponding Uniprot page. The Download, Reset, and Back buttons, respectively, provide users with the ability to download the data in a.csv format, reset filter conditions of all control widgets, and return to the Home interface. Furthermore, in the Home module, users can also access the 10 category results of the IVW model (the major analysis method), by checking the corresponding checkbox on the right-hand side (note: only one checkbox can be checked at a time). For example, checking the 'Disease' checkbox will allow users to jump to the results page containing 17,168 disease records.

## Study design module

The study design module provides users with a complete MR study design. Moreover, corresponding checkboxes are provided below to allow users to easily access the corresponding details (note: only one checkbox can be selected at a time), as illustrated in *Figure 2—figure supplement 2*.

## IVW interactive and IVW static modules

The IVW interactive module, as shown in *Figure 2—source data 1*, contains 21 control widgets, including the same search control widgets as in the Home module (due to a large amount of data, the initial values for Classification I and Classification II are set to 'Medical laboratory science' and 'blood lipids and lipoproteins,' respectively; users can reset the conditions as needed). In addition, the IVW interactive module includes the following new features: (i) a checkbox for 'Exposures with no effect,' which, when selected (checked by default), allows users to simultaneously remove all exposure traits that do not have a significant association (p>0.05) with any of the 16 outcome traits; (ii) a checkbox for '80 traits with consistent effect across three main outcome datasets,' which, when selected (unchecked by default), provides users with the option to filter results that have a significant and consistent causal association with all the three main outcome traits of AD (p<0.05); (iii) Download Data and Download Interactive buttons provide users with the ability to download data in a.csv format and images in a.html format, respectively. The IVW interactive module also provides users with interactive visual results (*Figure 2—figure supplement 3*). Clicking on the dots on the image will display detailed information on the corresponding id.exposure, exposure, id.outcome, outcome, beta, OR_CI, pval, -log10(pval), nsnp, method, Heterogeneity test, and MR-Egger intercept test. The IVW static module is the same as the IVW interactive module in terms of all the control widgets except for the absence of the Download Interactive button, and contains only static graph results.

## Sensitivity analysis interactive and sensitivity analysis static modules

The sensitivity analysis interactive module contains 21 control widgets, all of which are the same as those in the IVW interactive module, except that the '80 traits with consistent effects across three main outcome datasets' checkbox is absent, and that the id.outcome option is only available for single selection (initial value: ieu-b-2, users can reset the conditions as needed). In addition, a sensitivity analysis passed select box widget has been added, which allows users to select one or more of the six sensitivity analysis models and obtain statistically significant results (p<0.05), as shown in *Figure 2—source data 2*. Interactive visual results are also provided to users. By clicking on the squares in the image, detailed information for id.exposure, exposure, id.outcome, outcome, beta, OR_CI, pval,

nsnp, and method is displayed, as shown in *Figure 2—figure supplement 4*. The sensitivity analysis static module has the same control widgets as the Sensitivity Analysis Interactive module, except that it does not include the Download Interactive button and only contains static image results.

In our view, as the first interactive comprehensive platform for AD MR research to date, this online platform would benefit the field of scientific research in AD in numerous ways. On the one hand, it would allow researchers to quickly identify risk or protective factors from their own research and generate novel hypothesis regarding the molecular mechanism of AD. On the other hand, it would allow researchers with complementary expertise to provide multiple characterizations of the same data. As the platform is hosted on a server and accessed through a web interface, which could meet the multi-terminal compatibility, thereby MRAD's online presence could increase access to potential users.

## MRAD utility data mining

To demonstrate the utility of the MRAD platform, we focus on the IVW model-identified exposure traits that have significant and consistent effects across three main outcome traits of AD to demonstrate the performance of the MRAD platform. Detailed investigation and reporting of other factors will be carried out in future research.

In this study, MR analysis was first performed on the three main outcome traits of AD to explore their potential risk or protective factors, leading to the identification of a total of 80 exposure traits (p<0.05), which fell into five Classification I categories: Medical laboratory science (n=51), Family history (n=10), Disease (n=9), Molecular trait (n=7), and Lifestyle trait (n=3). A total of 63 exposure traits (risk factors) were positively associated with all the three main outcome traits, while 16 exposure traits (protective factors) were negatively associated with the three main outcome traits, with Ulcerative colitis (ebi-a-GCST000964) being negatively associated with the AD outcome traits of ieu-b-2 and ieu-a-297, and positively associated with the AD outcome traits of ieu-b-5067. MR analysis was performed on the outcome traits of 13 different AD-finn subtypes to further examine the causal association between the above-identified key common exposure traits and different subtypes of AD outcome traits. The results are provided below in detail.

## Causal association between medical laboratory science and the main outcome traits of AD

In this study, the 51 medical laboratory science items that each had a causal effect on the main outcome traits of AD were grouped into three Classification II categories (blood lipids and lipoproteins (n=36), immunological tests (n=12), and plasma protein tests (n=3)).

### Blood lipids and lipoproteins

A total of 36 blood lipids and lipoproteins items as exposure traits had effects on the main outcome traits of AD: (1) 32 of which were positively associated with the main outcome traits, seven of which, e.g., apolipoprotein B (ieu-b-108), were positively associated with EOAD (finn-b-AD_EO) and LOAD (finn-b-AD_LO); free cholesterol in IDL (met-c-868) was positively associated with EOAD (finn-b-AD_EO); four of which, e.g., phospholipids in small LDL (met-d-S_LDL_PL), were positively associated with LOAD (finn-b-AD_LO), as shown in *Figure 3A*. The corresponding sensitivity analysis and Bonferroni correction results are shown in *Figure 3—figure supplement 1* and *Supplementary file 1*. (2) four of which were negatively associated with the main outcome traits, apolipoprotein A-I (ieu-b-107) was negatively associated with both EOAD (finn-b-AD_EO) and LOAD (finn-b-AD_LO), and the negative causal association was slightly stronger for EOAD than for LOAD; phospholipids to total lipids ratio in chylomicrons and extremely large VLDL (met-d-XXL_VLDL_PL_pct) was negatively associated with LOAD (finn-b-AD_LO). These findings are illustrated in *Figure 3B*. The corresponding sensitivity analysis and Bonferroni correction results are shown in *Figure 3—figure supplement 2* and *Supplementary file 1*.

### Immunological tests

A total of 12 immunological test items as exposure traits had positive effects on the main outcome traits of AD. Six of which, e.g., CD33 on Monocytic Myeloid-Derived Suppressor Cells (ebi-a-GCST90001952), were positively associated with LOAD (finn-b-AD_LO), as shown in *Figure 3C*. The

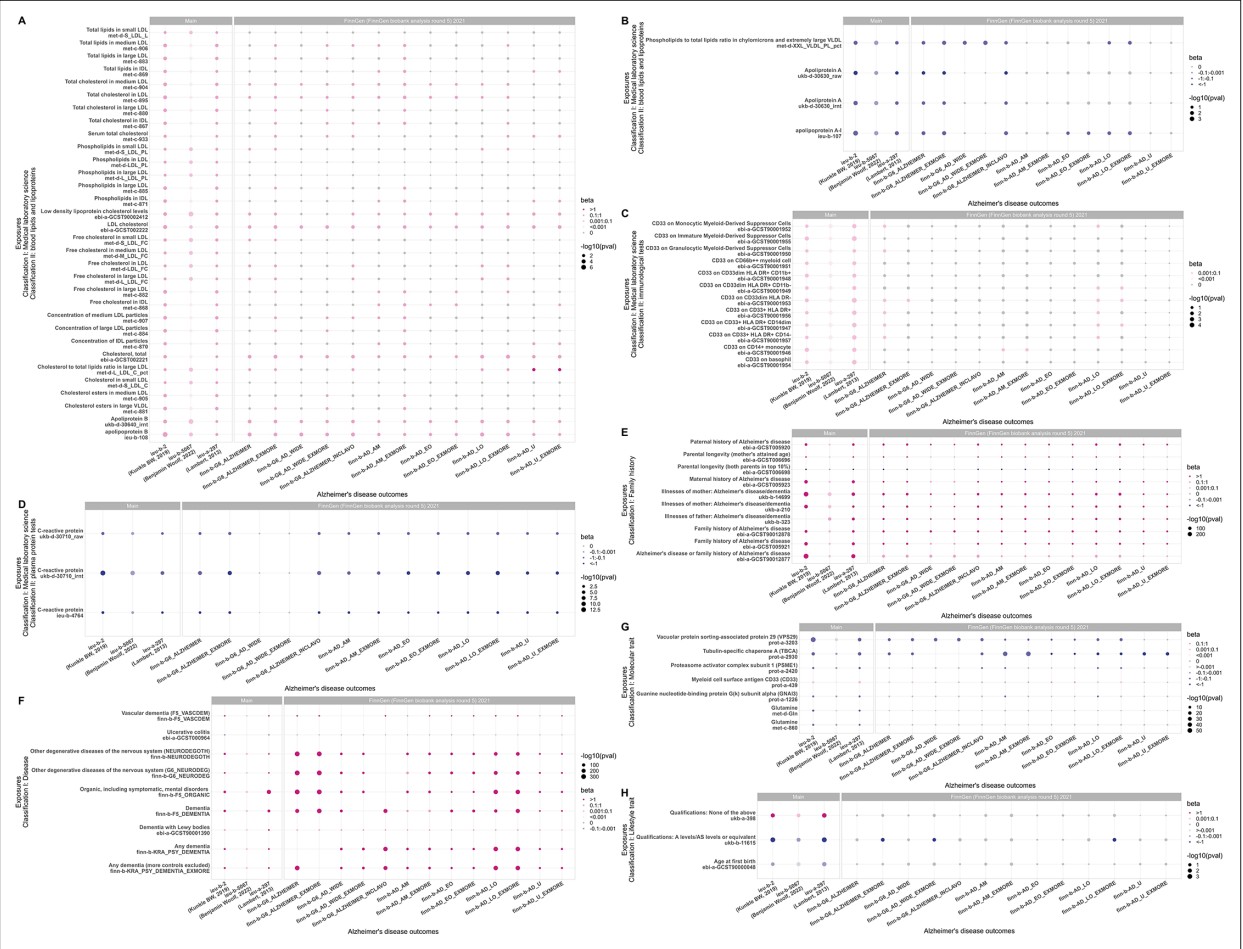

**Figure 3.** 80 exposure traits with causal effects on the main outcome traits of Alzheimer's disease (AD) based on the major analysis method random-effects inverse-variance weighted (IVW) model. (**A**) Thirty-two blood lipids and lipoproteins items that were positively associated with the main outcome traits of AD. (**B**) Four blood lipids and lipoproteins items that were negatively associated with the main outcome traits of AD. (**C**) Twelve immunological test items that were positively associated with the main outcome traits of AD. (**D**) Three plasma protein test items that were negatively associated with the main outcome traits of AD. (**E**) Ten family history items with causal effects on the main outcome traits of AD. (**F**) Nine disease items with causal effects on the main outcome traits of AD. (**G**). Seven molecular trait items with causal effects on the main outcome traits of AD. (**H**) Three lifestyle trait items with causal effects on the main outcome traits of AD. Note: The pink dots in the figure represent positive association, the blue dots in the figure represent negative association, with the color depth of the dots being positively proportional to the OR value (the darker the color, the larger the OR value), and the size of the dots being inversely proportional to the p-value (the smaller the p-value, the larger the dots). The gray dots represent no significant causal association (p>0.05).

The online version of this article includes the following figure supplement(s) for figure 3:

**Figure supplement 1.** Statistical models for causal effect results of thirty-two blood lipids and lipoproteins items that were positively associated with the main outcome traits of Alzheimer's disease (AD).

**Figure supplement 2.** Statistical models for causal effect results of four blood lipids and lipoproteins items that were negatively associated with the main outcome traits of Alzheimer's disease (AD).

**Figure supplement 3.** Statistical models for causal effect results of twelve immunological test items that were positively associated with the main outcome traits of Alzheimer's disease (AD).

**Figure supplement 4.** Statistical models for causal effect results of three plasma protein tests items that were negatively associated with the main outcome traits of Alzheimer's disease (AD).

**Figure supplement 5.** Statistical models for causal effect results of ten family history items with causal effects on the main outcome traits of Alzheimer's disease (AD).

**Figure supplement 6.** Statistical models for causal effect results of nine disease items with causal effects on the main outcome traits of Alzheimer's disease (AD).

*Figure 3 continued on next page*

*Figure 3 continued*

**Figure supplement 7.** Statistical models for causal effect results of seven molecular trait items with causal effects on the main outcome traits of Alzheimer's disease (AD).

**Figure supplement 8.** Statistical models for causal effect results of three lifestyle trait items with causal effects on the main outcome traits of Alzheimer's disease (AD).

corresponding sensitivity analysis and Bonferroni correction results are shown in *Figure 3—figure supplement 3* and *Supplementary file 1*.

### Plasma protein tests

A total of three plasma protein test items as exposure traits had negative effects on the main outcome traits of AD. The three exposure traits were C-reactive protein (ukb-d-30710_raw, ukb-d-30710_irnt, and ieu-b-4764). All the three exposure traits were negatively associated with EOAD (finn-b-AD_EO) and LOAD (finn-b-AD_LO), as shown in *Figure 3D*. The corresponding sensitivity analysis and Bonferroni correction results are shown in *Figure 3—figure supplement 4* and *Supplementary file 1*.

## Causal association between family history and the main outcome traits of AD

A total of 10 family history items as exposure traits had causal effects on the main outcome traits of AD. In particular, a parental or family history of AD increased the overall risk of developing AD, and was positively associated with both EOAD (finn-b-AD_EO) and LOAD (finn-b-AD_LO), as shown in *Figure 3E*. The corresponding sensitivity analysis and Bonferroni correction results are shown in *Figure 3—figure supplement 5* and *Supplementary file 1*.

## Causal association between diseases and the main outcome traits of AD

In this study, the nine disease items that each had a causal effect on the main outcome traits of AD were grouped into four Classification II categories (dementia (n=5), neurodegenerative diseases (n=2), mental disorders associated with neurological diseases (n=1), and digestive system diseases (n=1)). Their causal effects with the main outcome traits of AD and the outcome traits of EOAD (finn-b-AD_EO) and LOAD (finn-b-AD_LO) are shown in *Figure 3F*. The corresponding sensitivity analysis and Bonferroni correction results are shown in *Figure 3—figure supplement 6* and *Supplementary file 1*.

## Causal association of molecular traits with the main outcome traits of AD

A total of seven molecular trait items as exposure traits had causal effects on the main outcome traits of AD, among which Myeloid cell surface antigen CD33 (prot-a-439) was positively associated with the main outcome traits of AD, as well as with both EOAD (finn-b-AD_EO) and LOAD (finn-b-AD_LO). The remaining six were all negatively associated with the main outcome traits of AD, and their causal effects on the outcome traits of 13 AD-finn subtypes were as follows: (i) tubulin-specific chaperone A (TBCA; prot-a-2930) and vacuolar protein sorting-associated protein 29 (VPS29; prot-a-3203) were negatively associated with both EOAD (finn-b-AD_EO) and LOAD (finn-b-AD_LO); (ii) guanine nucleotide-binding protein G(k) subunit alpha (GNAI3; prot-a-1226) and proteasome activator complex subunit 1 (PSME1; prot-a-2420) were negatively associated with LOAD (finn-b-AD_LO), but had no significant causal association with EOAD (finn-b-AD_EO) ($p>0.05$); and (iii) neither glutamine (met-c-860) nor glutamine (met-d-Gln) had significant causal association with EOAD (finn-b-AD_EO) or LOAD (finn-b-AD_EO) ($p>0.05$), as shown in *Figure 3G*. The corresponding sensitivity analysis and Bonferroni correction results are shown in *Figure 3—figure supplement 7* and *Supplementary file 1*.

## Causal association of lifestyle traits with the main outcome traits of AD

A total of three lifestyle trait items as exposure traits had causal effects on the main outcome traits of AD. Their causal effects with the main outcome traits of AD and the outcome traits of EOAD (finn-b-AD_EO) and LOAD (finn-b-AD_LO) are shown in *Figure 3H*. The corresponding sensitivity analysis

and Bonferroni correction results are shown in *Figure 3—figure supplement 8* and *Supplementary file 1*.

## Discussion

Despite decades of research on AD, controversy still remains regarding which factors play an important in its pathogenesis. This study carried out hypothesis-free Mendelian randomization analysis for Alzheimer's disease, which provided a thorough and comprehensive evaluation with regard to risk or protective factors for AD. This MR study covers most exposure traits that are causally associated with AD outcome traits, including diseases, medical laboratory science items, imaging items, anthropometric items, treatments, molecular traits, gut microbiota, past histories, family histories, and lifestyle traits, and reveals the causal associations between these exposure traits and different AD subtypes.

Based on this, for the convenience of display and operation, a user-friendly prediction platform was built online called MRAD. The MRAD provides a one-stop online analysis service for researchers worldwide, including data retrieval → visualization → personalized analysis → data download. Users can obtain analysis results of different MR models (the main IVW model and six sensitivity analysis models) on 18,097 exposure traits and 16 AD outcome traits, totaling 400,274 records, and are allowed to set personalized parameters to meet different analysis needs. Additionally, the MRAD provides interactive visualization interfaces and download functions for the above results.

MRAD platform provides a unique resource for systematically identifying risk or protective factors of AD, which facilitates early identification, diagnosis, prevention, and treatment, with significant clinical and social value. It could have several strengths: (i) The current methods for identifying AD mainly rely on assessment scales, cerebrospinal fluid (CSF) examinations, and brain PET/MRI. However, assessment scales can be biased by factors such as the anxiety and nervousness of the subjects. CSF examinations require an invasive lumbar puncture, leading to low patient acceptance. PET/MRI scans are expensive and have limited equipment accessibility. These limitations restrict early AD identification. Thus, there is a pressing clinical need for readily available, time- and cost-effective, and accurate detection methods. In this study, the Medical laboratory science and Molecular trait used could be less expensive, faster to detect, easier to operate, and more accessible for widespread adoption. They hold great value for early AD identification and have the potential to become crucial tools for identifying AD in the future. (ii) Imaging acts as a powerful assistive tool for diagnosing Alzheimer's disease. Traditional imaging examinations mainly depict changes in the brain's macroscopic structure, while research on microstructural changes in disease-related areas is relatively limited. Studies have demonstrated that microstructural neurodegenerative processes are extensive and pronounced during AD progression. Our study results cover traditional macroscopic neuroimaging results and reveal numerous potential causal relationships between brain microstructure and AD. The combination of macroscopic and microstructural insights will provide more valuable information for clinical diagnosis. (iii) Clarifying the patient's disease, past history, and family history can aid in preventing AD at an early stage, and prevention of AD could be attained through monitoring anthropometric indicators, improving gut microbiota, and adjusting lifestyle traits. (iv) Currently, the development of new drugs for AD is mainly underscored by Aβ, Tau, and other inhibitors. Since 2000, global pharmaceutical companies have invested hundreds of billions of dollars in the development of new drugs for AD, and these drugs have not yielded successful results. AD drug development has thus been perceived as having the highest failure rate of all drug research, reaching 99.6%. Hence, further research on molecular traits to find new targets and develop new drugs for these targets will provide new pathways for AD treatment.

To briefly demonstrate the performance of MRAD, we explored the IVW model-identified exposure traits that had significantl consistentl effect across all the three main outcome traits of AD.

The association of lipids and lipoproteins, C-reactive protein, family histories, neurological disorders, glutamine, and education level with AD has been widely reported (*Hu et al., 2022*; *Picard et al., 2022*; *Caramelli et al., 1999*; *Kuo et al., 1998*; *Raygani et al., 2006*; *Takechi et al., 2009*; *Takechi et al., 2010*; *Liu et al., 2020*; *Wu et al., 2019*; *Wang et al., 2020*; *Zhou et al., 2020*; *Nordestgaard et al., 2022*; *Wingo et al., 2022*; *Hartmann, 2001*; *Tynkkynen et al., 2018*; *Hashemi et al., 2022*; *Treiber-Held et al., 2003*; *Zarrouk et al., 2018*; *Zuin et al., 2021*; *Tong et al., 2022*; *Button et al., 2019*; *Cannon-Albright et al., 2019*; *Wang et al., 2019*; *Letenneur et al., 1994*; *Stern et al., 1994*) and is consistent with the results of this study. Moreover, given that the prevalence of LOAD is about

95% in patients with AD and that LOAD has a stronger genetic predisposition than EOAD (*Godyń et al., 2016*; *Ransohoff and El Khoury, 2015*; *Zou et al., 2014*), identifying new risk genes for LOAD is crucial for understanding its potential etiology. Therefore, this study further explored the relationships between these traits and different AD subtypes, leading to the following findings: (i) apolipoprotein B, cholesterol, total, LDL cholesterol, Low-density lipoprotein cholesterol levels, total cholesterol in LDL, total cholesterol in medium LDL, cholesterol to total lipids ratio in large LDL, free cholesterol in large LDL, free cholesterol in LDL, phospholipids in small LDL, parental or family history of AD, parental longevity (mother's attained age), dementia, vascular dementia, dementia with Lewy bodies, other degenerative diseases of the nervous system, and organic, including symptomatic, mental disorders were all positively associated with LOAD; (ii) apolipoprotein A-I, phospholipids to total lipids ratio in chylomicrons and extremely large VLDL, C-reactive protein, parental longevity (both parents in top 10%), and qualifications: A levels/AS levels or equivalent were all negatively associated with LOAD. These findings suggest that the above traits may have critical impacts on LOAD.

Moreover, some novel potential therapeutic targets of AD were identified as follows: CD33 on Monocytic Myeloid-Derived Suppressor Cells, CD33 on CD33 + HLA DR +CD14 dim, CD33 on CD33 + HLA DR+, CD33 on CD33 + HLA DR + CD14-, CD33 on CD33dim HLA DR -, CD33 on CD33dim HLA DR + CD11b-, and Myeloid cell surface antigen CD33 were positively associated with all the three main outcome traits of AD and the risk of LOAD. It has been reported that CD33 is a 67 kDa glycosylated transmembrane protein, a member of the sialic acid-binding immunoglobulin-like lectins family (SIGLECS family), which is an important receptor for cell growth and survival, as well as a critical receptor for the clathrin-independent endocytosis pathway and the innate and adaptive immune system functions. CD33 is mainly expressed in microglia, which are a type of glial cells in the central nervous system (*Villegas-Llerena et al., 2016*). Meanwhile, the splicing efficiency of CD33 affects microglia activation (*Malik et al., 2013*). Several genome-wide association studies have demonstrated that CD33 is a high-risk gene for AD (*Hollingworth et al., 2011*; *Gu et al., 2022*). In animal models, knockdown of CD33 significantly reduced amyloid plaque levels and knockout mice did not exhibit other health defects. Sialylated glycoproteins and glycolipids on amyloid plaques bind to CD33, which is most likely the cause of the amyloid 'immune escape' (*Griciuc et al., 2013*). Furthermore, polymorphisms in CD33 can increase the risk of AD by causing neuronal degeneration in the hippocampal and parahippocampal regions of the brain (*Wang et al., 2017*). Downregulation of the sialic acid-binding domain of CD33 can reduce the risk of developing AD. Therefore, inhibiting CD33 is an effective approach to inhibiting the development of AD, and the sialic acid-binding site on CD33 is a promising pharmacophore (*Miles et al., 2019*).

Tubulin-specific chaperone A (TBCA) was negatively associated with all the three main outcome traits of AD, as well as EOAD and LOAD (pval is significant at the Bonferroni threshold). TBCA is an important member of the tubulin-specific chaperones (TBCs) family. *Nolasco et al., 2005* demonstrated that TBCA can regulate the proportion of α and β-tubulin, enabling them to correctly aggregate into cellular microtubules. Cellular microtubules play important roles in many biological functions, especially in cell movement, cell division, intracellular transport, and cell structure. After silencing TBCA, abnormal microtubule aggregation occurs in mammalian cells, and the cells cannot grow and divide normally, ultimately leading to apoptosis (*Gaitanos et al., 2004*; *Cormier et al., 2008*). Moreover, studies have shown that TBCA plays a crucial role in correct β-tubulin folding and α/β-tubulin heterodimer formation (*Bellmunt et al., 2009*). Protein misfolding can lead to many diseases, such as neurodegenerative diseases. Additionally, higher levels of TBCA are significantly associated with lower AD risk (*Hillary et al., 2022*). These findings suggest that TBCA may serve as a potential protective factor against AD.

Vacuolar protein sorting-associated protein 29 (VPS29) was negatively associated with all the three main outcome traits of AD, as well as EOAD and LOAD (pval is significant at the Bonferroni threshold). VPS29 is a component of the retromer complex and is highly expressed in the brain, heart, and kidneys, playing an essential role in retromer functions such as synaptic transmission, survival, and

movement (*Ye et al., 2020*). Retromer mainly consists of the VPS26-VPS29-VPS35 trimer and Sorting Nexins (SNXs), and its defects are closely related to various human diseases, including neurodegenerative diseases (*Ye et al., 2020*). Studies have reported that VPS29 knockdown leads to reduced levels of VPS35 and VPS26 (*Fuse et al., 2015*; *Jimenez-Orgaz et al., 2018*), which regulates the localization of retromer within neurons and is essential for the aging nervous system (*Ye et al., 2020*). The retromer complex has been found to regulate the transport of a variety of substances, including amyloid precursor protein (APP), β-secretase, and phagocytic receptors on microglia. The retromer complex regulates the production of amyloid-β (Aβ) by regulating the transport of relevant carrier proteins, thus playing a role in AD (*Zhang et al., 2016*). When the retromer complex malfunctions, the pathway for the reverse transport of APP and β-secretase to the trans-Golgi network is disrupted, resulting in an increase in the production of Aβ, which accelerates the pathological process of AD (*Seaman, 2021*). Meanwhile, the reduction of phagocytic receptors on the surface of microglia weakens the clearance and protective functions of microglia. Recent studies have shown that stabilizing the retromer complex through chaperone proteins can limit the amyloid processing of APP to reduce the production of Aβ (*Zhang et al., 2016*). These findings suggest that the retromer complex can serve as a new therapeutic target to intervene in the pathological progression of AD.

Guanine nucleotide-binding protein G(k) subunit alpha (GNAI3) was negatively associated with the three main outcome traits of AD and the risk of LOAD. G proteins are a class of signal transduction proteins that can bind with guanosine diphosphate (GDP) and have guanosine triphosphate (GTP) hydrolysis activity; they have more than 40 types, consisting of alpha, beta, and gamma subunits with a total molecular weight of about 100 kDa, with the alpha subunit having the greatest variation and determining the specificity of the G proteins (*Heese, 2013*). G proteins are intracellular membrane proteins that shuttle between receptors and effector proteins, acting as signal transducers and playing an absolute dominant role in transmembrane cell signaling in the body. All cellular activities are related to signals, and signals are the initiating factors of all cell activities, while physiological responses are only the final results of signals acting on cells. After receiving external stimuli, cells respond by implementing signal transduction through a set of specific mechanisms to ultimately regulate the expression of specific genes, and the whole process is referred to as a cellular signaling pathway. In the pathogenesis of AD, the abnormal content and distribution of multiple signaling molecules, as well as the abnormality of signa transmission pathways, play an important role in AD pathological changes (*Fowler et al., 1995*), suggesting that gaining insights into signal transduction mechanisms may provide a potential new pathway to explore the pathogenesis of AD.

Proteasome activator complex subunit 1 (PSME1) was negatively associated with all the three main outcome traits of AD and the risk of LOAD. PSME1 is the encoding gene of the 11 s proteasome activator subunit (also known as PA28α) and is located on human chromosome 14q11.2. PA28α is an activator of the proteasome, which mainly increases the protein degradation activity of 20 S proteasome and participates in MHC-I (major histocompatibility complex I) restricted antigen presentation (*Adelöf et al., 2018*). Studies have shown that PA28α overexpression in the brain of female mice can effectively prevent protein aggregation in the hippocampus, thereby reducing depression-like behavior and enhancing learning and memory ability (*Donggui, 2021*). Related studies have shown that proteasome function and PA28α expression are inhibited in the brains of diabetic rats (*Donggui, 2021*). The PA28 expression in the diabetic brain has a certain regulatory effect on protein metabolism caused by oxidative damage (*Donggui, 2021*). As suggested above, PSME1 may be a new potential therapeutic target for AD and deserves further investigation.

In conclusion, this is one of the most comprehensive studies to provide important insight into the genetic etiology underlying AD based on hypothesis-free Mendelian randomization analysis. In the meantime, we developed the first MR platform for AD, of great clinical and scientific significance that provided a thorough and comprehensive evaluation with regard to risk or protective factors for AD. It also provided physicians and scientists with a very convenient, free as well as user-friendly tool for further scientific investigation. It is important to notice that we recognized CD33, TBCA, VPS29, GNAI3, and PSME1 as novel potential therapeutic targets for AD that deserve further investigation in

more detail. However, in this study, since the GWAS datasets for both the exposure and the outcome traits (AD) selected were obtained from the public database (MRC IEU OpenGWAS), where the GWAS datasets for AD are only of European population, and since we use the TwoSampleMR, which requires that the populations for the exposure traits and the outcome traits be the same to satisfy the requirement for a control variable, this study currently has certain limitations in terms of population. We initiated a Mendelian randomization study on AD at clinical hospitals in China and are currently in the sample collection stage to address the limitations. In the future, we will integrate data from more populations and continuously update new advances in AD research to explore its potential differences in different populations.

# Materials and methods

**Key resources table**

| Reagent type (species) or resource | Designation | Source or reference | Identifiers | Additional information |
|---|---|---|---|---|
| Software, algorithm | MRC IEU OpenGWAS | *Elsworth et al., 2020* | https://gwas.mrcieu.ac.uk/ | |
| Software, algorithm | UniProt | *UniProt Consortium, 2018* | https://www.uniprot.org/ RRID:SCR_002380 | |
| Software, algorithm | EVenn | *Tong et al., 2022* | http://www.ehbio.com/test/venn/#/ | |
| Software, algorithm | R (version 4.1.2) software | Simon Urbanek et al. | https://www.r-project.org/ RRID:SCR_001905 | |
| Software, algorithm | TwoSampleMR (version 0.5.8) | *Hemani et al., 2018* | https://github.com/MRCIEU/TwoSampleMR RRID:SCR_019010 | TwoSampleMR package in the R (version 4.1.2) software |
| Software, algorithm | Shiny (version 1.8.0) | rstudio | https://shiny.posit.co/ RRID:SCR_001626 | Shiny package in R (version 4.1.2) |
| Software, algorithm | MRAD | this paper | https://gwasmrad.com/mrad/ | This paper, a new online data analysis platform for identifying the risk or protective factors for AD called MRAD. |

## Database and software

The following databases and software packages were used in this study: MRC IEU OpenGWAS (*Elsworth et al., 2020*; https://gwas.mrcieu.ac.uk/), UniProt (*UniProt Consortium, 2018*; https://www.uniprot.org/), EVenn (*Chen et al., 2021*; http://www.ehbio.com/test/venn/#/), R (version 4.1.2) software (*R Development Core Team, 2021*).

## MR design for AD (Figure 4)

### Data sources

#### Exposure traits

Inclusion criteria: datasets of the European population.

Exclusion criteria: (i) eQTL-related datasets; (ii) AD-related datasets.

In this study, the GWAS datasets selected were derived from 42,335 GWAS datasets in the public database (MRC IEU OpenGWAS, https://gwas.mrcieu.ac.uk/). Based on the above inclusion and exclusion criteria, 19,942 eQTL-related datasets were excluded first, leaving 22,393 GWAS datasets. Next, the datasets with the European population were selected, and 18,117 GWAS datasets were obtained. Finally, 20 AD-related datasets were excluded; 18,097 GWAS datasets were obtained at the end as the exposure traits of this study (See *Supplementary file 2* for basic information).

#### Outcome traits

Inclusion criteria: (i) datasets of patients with AD with complete information and clear data sources; (ii) datasets of the European population.

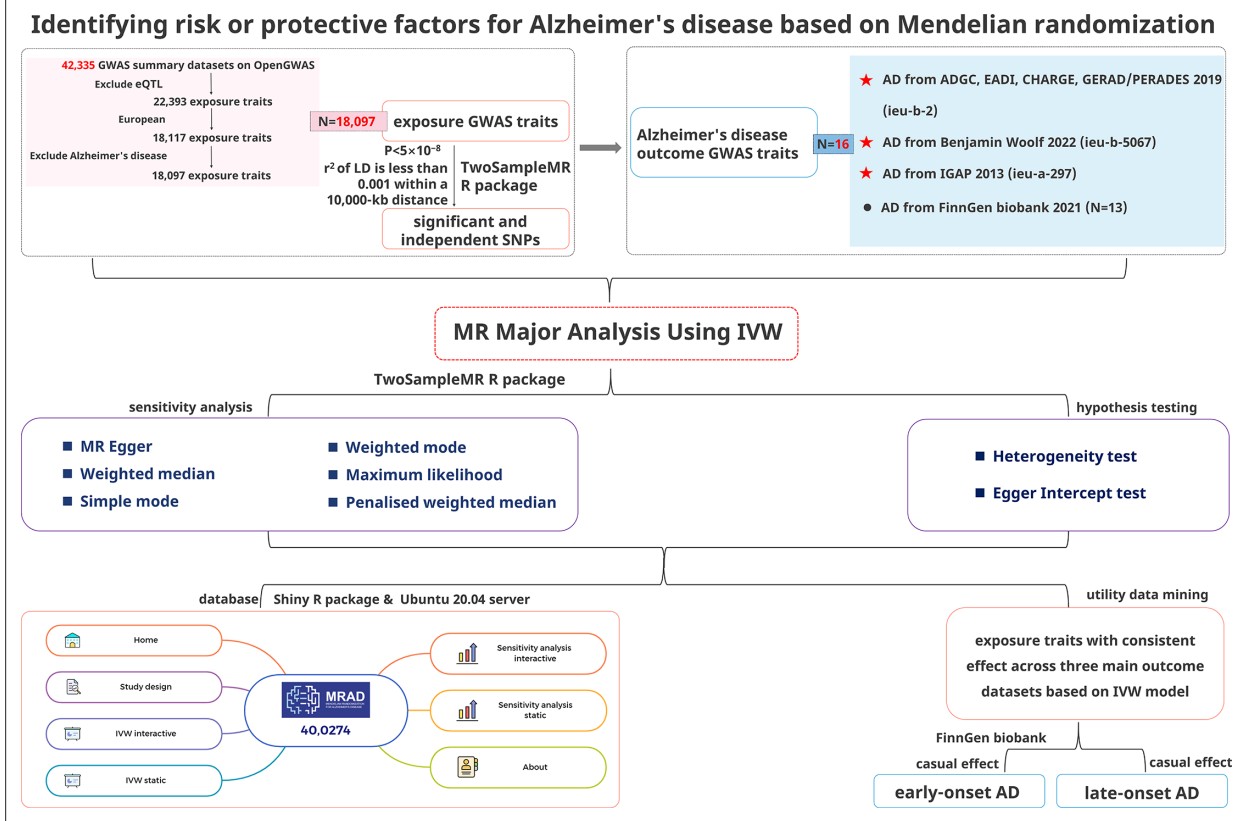

**Figure 4.** Study design.

Exclusion criteria: (i) Number of SNPs <1 million; (ii) datasets with unspecified sex; (iii) datasets with a family history of AD; (iv) datasets with dementia.

Based on the above criteria, 16 GWAS datasets of outcome traits were selected from the MRC IEU OpenGWAS database, comprising datasets of AD from Alzheimer Disease Genetics Consortium (ADGC), Cohorts for Heart and Aging Research in Genomic Epidemiology Consortium (CHARGE), The European Alzheimer's Disease Initiative (EADI), and Genetic and Environmental Risk in AD/Defining Genetic, Polygenic and Environmental Risk for Alzheimer's Disease Consortium (GERAD/PERADES) 2019 (ieu-b-2); AD from Benjamin Woolf 2022 (ieu-b-5067); AD from International Genomics of Alzheimer's Project (IGAP) 2013 (ieu-a-297) as the datasets of main outcome traits for AD, as well as 13 datasets from FinnGen biobank 2021 corresponding to various AD subtypes, referred to as AD-finn subtypes (as shown in *Figure 5*).

**Figure 5.** Basic information in 16 outcome traits in MRC IEU OpenGWAS.

## Selection of instrumental variables

SNPs serve as instrumental variables for MR research. In this study, 18,097 exposures-variable SNPs were selected for MR research from the GWAS data (as mentioned in *Exposure traits*), respectively, with the selected SNPs fulfilling the following requirements: (i) a genome-wide significant association with risk factors ($p<5 \times 10^{-8}$) in the European 1000 Genomes Project reference panel; (ii) independent of one another (that is, the r2 of linkage disequilibrium (LD) is less than 0.001 within a 10,000 kb distance) to avoid potential biases caused by LD between SNPs in the analysis.

## Statistical models for causal effect inference

A random-effects IVW model was used in this study as the major analysis method to uncover potential risk or protective factors for AD. The random-effects IVW model is the gold standard for MR studies, its principle is to calculate the inverse of the variance of each IV as its weight, assuming all IVs are valid. The regression does not include an intercept term, and the final result is the weighted average of the effect estimates from all IVs (*Bowden et al., 2017*). This model indicates that the true effect values may vary across different studies due to both sampling error and the heterogeneity of the true effect. The weight of each study is jointly determined by its inverse variance and the estimated heterogeneity variance. Thus, as long as there is no pleiotropy, even when there is significant heterogeneity ($p<0.05$), this method is preferred.

To assess the robustness of the IVW results, sensitivity analysis was performed using six additional models: (i) MR-Egger: MR-Egger's biggest difference from IVW is that it considers the intercept term during regression to evaluate bias caused by horizontal pleiotropy. The intercept represents the magnitude of horizontal pleiotropy, with a value close to 0 indicating minimal pleiotropy. The primary purpose is to detect and correct for horizontal pleiotropy. Thus, when significant horizontal pleiotropy is observed ($p<0.05$), this method is preferred (*Burgess and Thompson, 2017*; *Slob et al., 2017*). (ii) Weighted median: The weighted median method is a technique for evaluating causal relationships using a majority of genetic variants (SNPs). If at least 50% of the SNPs are valid IVs, the median of the causal estimates will tend toward the true causal effect. This method provides an unbiased estimate (i.e. the 'majority validity' assumption) (*Bowden et al., 2016*). (iii) Simple mode: Involves comparing the frequencies or proportions of genotypes or phenotypes between control and experimental groups. Moreover, it can illustrate whether the observed differences in genotypes or phenotypes between the two groups are statistically significant. (iv) Weighted mode: The weighted mode method is a technique for combining multiple Mendelian randomization estimates. This method assigns weights to the causal effect estimates of different genetic variants on the trait and then takes the weighted mode as the final estimate of the causal effect. In genetic variant estimates, the method can decrease bias caused by outliers. (v) Maximum likelihood: This method is used when it is known that a random sample follows a particular probability distribution; however, the specific parameters of that distribution remain unknown, and it involves conducting multiple experiments, observing the results, and using those results to infer the approximate values of the parameters (*Xue et al., 2021*). (vi) Penalized weighted median: An enhanced version of the weighted median estimate that provides a consistent estimate of the causal effect. (vii) Heterogeneity and horizontal pleiotropy assessment use the heterogeneity tests (*Higgins et al., 2003*) and Egger intercept tests (*Bowden et al., 2015*), respectively.

The above analyses were performed using the TwoSampleMR (*Hemani et al., 2018*) package in the R (version 4.1.2) software. Association of exposures with outcomes was assessed using odds ratio (OR) and 95% confidence interval (95% CI), with OR >1 indicating a positive association (risk factor) and $0<OR<1$ indicating a negative association (protective factor). Differences with a two-sided $p<.05$ were considered statistically significant. Furthermore, owing to the relatively large number of exposure and outcome traits included in this study, the multiple testing correction method Bonferroni correction was added to identify significant hits, the threshold for Bonferroni-corrected was 0.05 divided by 289,552 tests ($p<1.727e-07$).

## Building the MRAD platform

In this study, the online MRAD platform was developed using the Shiny package *Shiny, 2024* in R (version 4.1.2) and hosted on an Ubuntu 20.04 server. By leveraging Shiny, we combined the

computational capabilities of R with modern web technologies, allowing to us construct an interactive user interface with novel approaches.

## Code availability

The MRAD platform can be freely accessed online at https://gwasmrad.com/mrad/. The main project development repository: https://github.com/ZhaoTianyu-zty/MRAD, (copy archived at *Zhao, 2023*).

## Acknowledgements

We would like to thank Taylor & Francis (https://www.tandfeditingservices.com) for English language editing.

## Additional information

### Funding

| Funder | Grant reference number | Author |
| --- | --- | --- |
| National Natural Science Foundation of China | 82302872 | Meishuang Zhang |
| Changchun Science and Technology Planning Project | 21ZY18 | Li Chen |

The funders had no role in study design, data collection and interpretation, or the decision to submit the work for publication.

### Author contributions

Tianyu Zhao, Data curation, Formal analysis, Visualization, Methodology, Writing - original draft; Hui Li, Software; Meishuang Zhang, Yang Xu, Ming Zhang, Writing - review and editing; Li Chen, Conceptualization, Supervision

### Author ORCIDs

Tianyu Zhao https://orcid.org/0009-0000-6835-6482
Li Chen https://orcid.org/0000-0002-9601-4903

Reviewer #1 (Public Review): https://doi.org/10.7554/eLife.96224.3.sa1
Reviewer #2 (Public Review): https://doi.org/10.7554/eLife.96224.3.sa2
Author response https://doi.org/10.7554/eLife.96224.3.sa3

## Additional files

### Supplementary files

- Supplementary file 1. Sensitivity analysis and Bonferroni correction results.
- Supplementary file 2. Basic information of exposure traits in MRC IEU OpenGWAS.
- MDAR checklist

### Data availability

Publicly available datasets were analyzed in this study. These data can be found here: MRC IEU OpenGWAS at https://gwas.mrcieu.ac.uk/, and UniProt at https://www.uniprot.org/, the database searches were completed on January 30, 2023. The MRAD platform can be freely accessed online at https://gwasmrad.com/mrad/. The main project development repository is at https://github.com/ZhaoTianyu-zty/MRAD (copy archived at *Zhao, 2023*).

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
