## [Editor Report · eLife assessment]

This **important** study introduces the MRAD database, an advancement in Alzheimer's disease research that provides a powerful tool for evaluating risk and protective factors through Mendelian randomization analysis. The evidence supporting the database's utility is **solid**, with findings backed by robust data, though addressing methodological concerns and ensuring more rigorous validation of associations would further strengthen its impact. This resource represents a significant leap forward in the field, offering unprecedented opportunities for researchers and clinicians to uncover key insights into Alzheimer's etiology, potentially revolutionizing how Alzheimer's research is approached and accelerating the discovery of new prevention strategies and treatments.

---

## [Referee Report · Reviewer #1 (Public Review)]

Summary:

An online database called MRAD has been developed to identify the risk or protective factors for AD.

Strengths:

This study is a very intriguing study of great clinical and scientific significance that provided a thorough and comprehensive evaluation with regard to risk or protective factors for AD. It also provided physicians and scientists with a very convenient, free as well as user-friendly tool for further scientific investigation.

Comments on revised version:

The authors have resolved all of my previous comments. It's a decent paper worth to be published in this field.

---

## [Referee Report · Reviewer #2 (Public Review)]

Summary:

This MR study by Zhao et al. provides a comprehensive hypothesis-free approach to identifying risk and protective factors causal to Alzheimer's Disease (AD).

Strengths:

The study employs a comprehensive, hypothesis-free approach, which is novel over traditional hypothesis-driven studies. Also, causal associations between risk/protective factors and AD were addressed using genetic instruments and analysis.

---

## [Author Response]

The following is the authors’ response to the original reviews.

**Public Reviews:**

**Reviewer #1 (Public Review):**
Summary:An online database called MRAD has been developed to identify the risk or protective factors for AD.Strengths:This study is a very intriguing study of great clinical and scientific significance that provided a thorough and comprehensive evaluation with regard to risk or protective factors for AD. It also provided physicians and scientists with a very convenient, free as well as user-friendly tool for further scientific investigation.

We thank the reviewer for the conclusion and positive comments.

Weaknesses:(1) Comment: The paper mentions that the MRAD database currently contains data only from European populations, with no mention of data from other populations or ethnicities. Given potential differences in Alzheimer's Disease (AD) across different populations, the limitations of the data should be emphasized in the discussion, along with plans to expand the database to include data from more racial and geographic regions.

Thank you for your valuable comment. Further information regarding the limitations of populations is provided in the Conclusions section (page 19).

The newly added text describing the limitations of populations is as follows:

“However, in this study, since the GWAS datasets for both the exposure and the outcome traits (AD) selected were obtained from the public database (MRC IEU OpenGWAS), where the GWAS datasets for AD are only of European population, and since we use the TwoSampleMR, which requires that the populations for the exposure trait and the outcome trait be the same to satisfy the requirement for a control variable, this study currently has certain limitations in terms of population. We initiated a Mendelian randomization study on AD at clinical hospitals in China and are currently in the sample collection stage to address the limitations. In the future, we will integrate data from more populations and keep updating new progresses in AD research to explore its potential differences in different populations.”

(2) Comment: Sufficient information should be provided to clarify the data sources, sample selection, and quality control methods used in the MRAD database. Readers may expect more detailed information about the data to ensure data reliability, representativeness, and research applicability.

Thank you for your helpful suggestion. We appreciate you taking time and making effort in reviewing our manuscript and thank you for your insightful comments. We agree that adding more details is essential to make the manuscript more reliability, representativeness, and research applicability.

The newly added text describing more detailed information about the data is as follows:

(1) Sufficient information about data sources and sample selection (in the Data sources section of Methods section, page 8):

“Exposure traits

Inclusion criteria: datasets of the European population.

Exclusion criteria: (i) eQTL-related datasets; (ii) AD-related datasets.

“In this study, the GWAS datasets selected were derived from 42,335 GWAS datasets in the public database (MRC IEU OpenGWAS, https://gwas.mrcieu.ac.uk/). Based on the above inclusion and exclusion criteria, 19,942 eQTL-related datasets were excluded first, leaving 22,393 GWAS datasets. Next, the datasets with the European population were selected, and 18,117 GWAS datasets were obtained. Finally, 20 AD-related datasets were excluded; 18,097 GWAS datasets were obtained at the end as the exposure traits of this study (See Table S1 for basic information).

Outcome traits

Inclusion criteria: (i) datasets of patients with AD with complete information and clear data sources; (ii) datasets of the European population.

Exclusion criteria: (i) Number of SNPs <1 million; (ii) datasets with unspecified sex; (iii) datasets with a family history of AD; (iv) datasets with dementia.

Based on the above criteria, 16 GWAS datasets of outcome traits were selected from the MRC IEU OpenGWAS database, comprising datasets of AD from Alzheimer Disease Genetics Consortium (ADGC), Cohorts for Heart and Aging Research in Genomic Epidemiology Consortium (CHARGE), The European Alzheimer’s Disease Initiative (EADI), and Genetic and Environmental Risk in AD/Defining Genetic, Polygenic and Environmental Risk for Alzheimer’s Disease Consortium (GERAD/PERADES) 2019 (ieu-b-2); AD from Benjamin Woolf 2022 (ieu-b-5067); AD from International Genomics of Alzheimer's Project (IGAP) 2013 (ieu-a-297) as the datasets of main outcome traits for AD, as well as 13 datasets from FinnGen biobank 2021 corresponding to various AD subtypes, referred to as AD-finn subtypes. (as shown in Figure 2).”

(2) Sufficient information about quality control methods in the Statistical models for causal effect inference section of Methods section, page 9-10:

“A random-effects IVW model was used in this study as the major analysis method to uncover potential risk or protective factors for AD. The random-effects IVW model as the gold standard for MR studies, its principle is to calculate the inverse of the variance of each IV as its weight, assuming all IVs are valid. The regression does not include an intercept term, and the final result is the weighted average of the effect estimates from all IVs [34]. This model indicates that the true effect values may vary across different studies due to both sampling error and the heterogeneity of the true effect. The weight of each study is jointly determined by its inverse variance and the estimated heterogeneity variance. Thus, as long as there is no pleiotropy, even when there is significant heterogeneity (p < 0.05), this method remains the best MR model.

To assess the robustness of the IVW results, sensitivity analysis was performed using six additional models: (i) MR-Egger: MR-Egger’s biggest difference from IVW is that it considers the intercept term during regression to evaluate bias caused by horizontal pleiotropy. The intercept represents the magnitude of horizontal pleiotropy, with a value close to 0 indicating minimal pleiotropy. The primary purpose is to detect and correct for horizontal pleiotropy. Thus, when significant horizontal pleiotropy is observed (p < 0.05), this method is preferred [35,36]. (ii) Weighted median: The weighted median method is a technique for evaluating causal relationships using a majority of genetic variants (SNPs). If at least 50% of the SNPs are valid IVs, the median of the causal estimates will tend toward the true causal effect. This method provides an unbiased estimate (i.e., the “majority validity” assumption) [37]. (iii) Simple mode: Involves comparing the frequencies or proportions of genotypes or phenotypes between control and experimental groups. Moreover, it can illustrate whether the observed differences in genotypes or phenotypes between the two groups are statistically significant. (iv) Weighted mode: The weighted mode method is a technique for combining multiple Mendelian randomization estimates. This method assigns weights to the causal effect estimates of different genetic variants on the trait and then takes the weighted mode as the final estimate of the causal effect. In genetic variant estimates, the method can decrease bias caused by outliers. (v) Maximum likelihood: This method is used when it is known that a random sample follows a particular probability distribution; however, the specific parameters of that distribution remain unknown, and it involves conducting multiple experiments, observing the results, and using those results to infer the approximate values of the parameters [38]. (vi) Penalized weighted median: An enhanced version of the weighted median estimate that provides a consistent estimate of the causal effect. (vii) Heterogeneity and horizontal pleiotropy assessment use the heterogeneity tests [39] and Egger intercept tests [40], respectively.”

(3) Comment: While the authors mention that the MRAD database offers interactive visualization interfaces, the paper lacks detailed information on how to interpret and understand these visual results. Guidelines on effectively using these visualization tools to help researchers better comprehend the data are essential.

Thank you very much for your feedback, as we believe that our manuscript has been improved substantially as a result of your input. Owing to space constraints, the MRAD database user guide is included in the Supplementary Material. Meanwhile, for better understanding, the subheading of the relevant content in the Supplementary Material has been revised to “MRAD User Guide” (see Supplementary Material for details, page 11). Furthermore, considering user-friendliness, the user guide has been integrated into the database and can be accessed directly from the homepage by clicking on the “User Guide” module.

(4) Comment: In the conclusion section of the paper, it is advisable to explicitly emphasize the practical applications and potential clinical significance of the MRAD database. The paper should articulate how MRAD can contribute to the early identification, diagnosis, prevention, and treatment of AD and its potential societal and clinical value more clearly.

Thank you for pointing this out. In the Discussion section of the revised manuscript, we have now added how MRAD can contribute to the early identification, diagnosis, prevention, and treatment of AD and its potential societal as well as clinical value. And we reorganized the structure of Discussion section to make the text easier to understand, which could be helpful to further clarify the significance of MRAD. (page 15)

The newly added text describing the practical applications and potential clinical significance of the MRAD database is as follows:

“(i) The current methods for identifying AD mainly rely on assessment scales, cerebrospinal fluid (CSF) examinations, and brain PET/MRI. However, assessment scales can be biased by factors such as the anxiety and nervousness of the subjects. CSF examinations require an invasive lumbar puncture, leading to low patient acceptance. PET/MRI scans are expensive and have limited equipment accessibility. These limitations restrict early AD identification. Thus, there is a pressing clinical need for readily available, time- and cost-effective, and accurate detection methods. In this study, the Medical laboratory science and Molecular trait used could be less expensive, faster to detect, easier to operate, and more accessible for widespread adoption. They hold great value for early AD identification and have the potential to become crucial tools for identifying AD in the future. (ii) Imaging acts as a powerful assistive tool for diagnosing Alzheimer’s disease. Traditional imaging examinations mainly depict changes in the brain’s macroscopic structure, while research on microstructural changes in disease-related areas is relatively limited. Studies have demonstrated that microstructural neurodegenerative processes are extensive and pronounced during AD progression. Our study results cover traditional macroscopic neuroimaging results and reveal numerous potential causal relationships between brain microstructure and AD. The combination of macroscopic and microstructural insights will provide more valuable information for clinical diagnosis. (iii) Clarifying patient’s disease, past history, and family history can aid in preventing AD at an early stage, and prevention of AD could be attained through monitoring anthropometric indicators, improving gut microbiota, and adjusting lifestyle traits. (iv) Currently, the development of new drugs for AD is mainly underscored by Aβ, Tau, and other inhibitors. Since 2000, global pharmaceutical companies have invested hundreds of billions of dollars in the development of new drugs for AD, and these drugs have not yielded successful results. AD drug development has thus been perceived as having the highest failure rate of all drug research, reaching 99.6%. Hence, further research on molecular traits to find new targets and develop new drugs for these targets will provide new pathways for AD treatment.”

(5) Comment: Grammar and Spelling Errors: There are several spelling and grammar errors in the paper. Referring to a scientific editing service is recommended.

We appreciate your comments and suggestions for improving our manuscript. We have now used a professional editing service offered by Taylor and Francis to revise the grammar and language, and we have obtained a certificate of proof, which is attached. Thank you for recognizing our research, we have tried our best to improve the quality of this paper to ensure that it meets the high standards required for publication in of journal elife.

**Reviewer #2 (Public Review):**
Summary:This MR study by Zhao et al. provides a comprehensive hypothesis-free approach to identifying risk and protective factors causal to Alzheimer's Disease (AD).Strengths:The study employs a comprehensive, hypothesis-free approach, which is novel over traditional hypothesis-driven studies. Also, causal associations between risk/protective factors and AD were addressed using genetic instruments and analysis.

We greatly appreciate the positive feedback regarding the overall quality of our work.

Major comments:(1) Comment: The authors used the inverse-variance weighted (IVW) model as the primary method and other MR methods (MR-Egger, weighted mean, etc.) for sensitivity analysis. However, each method has its own assumption, and IVW is only robust when pleiotropy and heterogeneity are not severe. Rather than using IVW imprudently across all associations, it would be more appropriate to choose the best MR method for each association based on heterogeneity/Egger intercept tests. This customized approach, based on tests of MR assumption violations, yields more stable and reliable results. For reference, please follow up on work by Milad et al. (EHJ - "Plasma lipids and risk of aortic valve stenosis: a Mendelian randomization study"). This study selected the best MR model for each association based on pleiotropy and heterogeneity tests. Given the large number of tests in this work, I suggest initially screening significant signals using IVW, as done, and then validating the results using multiple MR methods for those signals. It is common for MR estimates from different methods to vary significantly (with some being statistically significant and others not), and in such cases, the MR estimates from the best-fitted model should be trusted and highlighted.

Thank you for your professional comments. We agree that our description of the Statistical models for causal effect inference was not specific enough. Therefore, we have included a new text describing more details about each method’s assumption and supplied a predefined approach to select the best statistical estimation from these methods in the Statistical models for causal effect inference section of Methods section (page 9-10). However, we would like to clarify our analysis method. In this study, the main analysis method used is the IVW random effects model instead of the IVW fixed effects model. The IVW random effects model indicates that the true effect values of different studies may vary, including both sampling error and heterogeneity of the true effect. The weight of each study is jointly determined by its inverse variance and the estimated heterogeneity variance. Thus, as long as there is no pleiotropy, even when there is significant heterogeneity (p < 0.05), this method is still the best MR model. We would like to thank you again for your feedback, as we believe that our manuscript has been improved substantially as a result of your input.

The newly added text describing more details about each method’s assumption and the customized best-fitted model is as follows:

“Statistical models for causal effect inference

A random-effects IVW model was used in this study as the major analysis method to uncover potential risk or protective factors for AD. The random-effects IVW model as the gold standard for MR studies, its principle is to calculate the inverse of the variance of each IV as its weight, assuming all IVs are valid. The regression does not include an intercept term, and the final result is the weighted average of the effect estimates from all IVs [34]. This model indicates that the true effect values may vary across different studies due to both sampling error and the heterogeneity of the true effect. The weight of each study is jointly determined by its inverse variance and the estimated heterogeneity variance. Thus, as long as there is no pleiotropy, even when there is significant heterogeneity (p < 0.05), this method remains the best MR model.

To assess the robustness of the IVW results, sensitivity analysis was performed using six additional models: (i) MR-Egger: MR-Egger’s biggest difference from IVW is that it considers the intercept term during regression to evaluate bias caused by horizontal pleiotropy. The intercept represents the magnitude of horizontal pleiotropy, with a value close to 0 indicating minimal pleiotropy. The primary purpose is to detect and correct for horizontal pleiotropy. Thus, when significant horizontal pleiotropy is observed (p < 0.05), this method is preferred [35,36]. (ii) Weighted median: The weighted median method is a technique for evaluating causal relationships using a majority of genetic variants (SNPs). If at least 50% of the SNPs are valid IVs, the median of the causal estimates will tend toward the true causal effect. This method provides an unbiased estimate (i.e., the “majority validity” assumption) [37]. (iii) Simple mode: Involves comparing the frequencies or proportions of genotypes or phenotypes between control and experimental groups. Moreover, it can illustrate whether the observed differences in genotypes or phenotypes between the two groups are statistically significant. (iv) Weighted mode: The weighted mode method is a technique for combining multiple Mendelian randomization estimates. This method assigns weights to the causal effect estimates of different genetic variants on the trait and then takes the weighted mode as the final estimate of the causal effect. In genetic variant estimates, the method can decrease bias caused by outliers. (v) Maximum likelihood: This method is used when it is known that a random sample follows a particular probability distribution; however, the specific parameters of that distribution remain unknown, and it involves conducting multiple experiments, observing the results, and using those results to infer the approximate values of the parameters [38]. (vi) Penalized weighted median: An enhanced version of the weighted median estimate that provides a consistent estimate of the causal effect. (vii) Heterogeneity and horizontal pleiotropy assessment use the heterogeneity tests [39] and Egger intercept tests [40], respectively.”

(2) Comment: Lines 157-160 mentioned "But to date, AD has been reported as hypothesis-driven MR study based on a single factor, ignoring the potential role of a huge number of other risk factors. Also, due to the high degree of heterogeneity present in AD subtypes, which have different biological and genetic characteristics. Thus, the previous studies cannot offer a systematic and complete viewpoint.". This statement overlooks a similar study published in Molecular Psychiatry ("A Phenome-wide Association and Mendelian Randomization Study for Alzheimer's Disease: A Prospective Cohort Study of 502,493"), which rigorously assessed the effects of 4171 factors spanning 10 different categories on AD using observational analysis and MR. The authors should revise their statement on the novelty of their study type throughout the manuscript and discuss how their work differs from and potentially strengthens previous studies.

Thank you for directing us to this literature. We have read this article carefully. This study shares some similarities with our study but there are significant differences with regards to sample sources and research fields. The study, as mentioned by the reviewer, used the UKB database as its sample source, and analyzed the association between 10 categories (comprising 4,171 factors) and AD, which were sociodemographic, physical measures, lifestyle and environment, health conditions, mental health, medications and operations, cognitive function, sex-specific factors, employment, and early-life factors. However, the study revealed they are restricted by the available variables from the UKB database, which lead to variables such as air pollution, blood glucose measures and so on were not included. Conversely, our study used samples from the MRC IEU OpenGWAS database, the largest open GWAS database globally. Furthermore, our research focus differs, as we primarily investigate the causal relationship between the following 10 categories (comprising 18,097 traits) and AD, which were Disease, Medical laboratory science, Imaging, Anthropometric, Treatment, Molecular trait, Gut microbiota, Past history, Family history, and Lifestyle trait. Most importantly, we have established a database encompassing all MR analysis results, allowing researchers and clinicians worldwide to conveniently and rapidly retrieve AD-associated risk factors via an online open integrated platform (MRAD, https://gwasmrad.com/mrad/).We have now added a new text in the Background section (page 6-7) describing the differences and potential strengthens towards previous studies.

The newly added text describing the differences and novelty towards previous studies is as follows:

“Chen et al. [30] used MR analysis to reveal the causal relationship between AD and factors including sociodemographic and early life status. However, the study revealed they are restricted by the available variables from the UKB database, which lead to variables such as air pollution, blood glucose measures and so on were not included. And also, due to the high degree of heterogeneity present in AD subtypes, which have different biological and genetic characteristics. Thus, the previous studies cannot offer a systematic and complete viewpoint. Our study uses the MRC IEU OpenGWAS database as the sample source for MR analysis to address the aforementioned limitations. The MRC IEU OpenGWAS database, the largest open GWAS database globally, has compiled 42,335 GWAS summary datasets from sources such as the UK Biobank, FinnGen Biobank, and Biobank Japan. Analyzing large-scale datasets will break new ground for MR research on AD.

MR requires a combination of background knowledge in biology, computer science, software studies, and statistics, which often leads to a dilemma where biologists are not well-versed in computer and statistical fields, while computer science experts struggle to adopt a medical biology mindset. Consequently, the vast majority of available GWAS data have not been effectively utilized through MR. Therefore, the construction of a multi-level data platform specifically for AD based on MR analysis of massive GWAS data is of great strategic significance, and it will facilitate researchers and clinicians worldwide to conveniently and rapidly obtain risk factors that are causally associated with AD.”

Reference:

[30] Chen SD, Zhang W, Li YZ, et al. (2023). A Phenome-wide Association and Mendelian Randomization Study for Alzheimer's Disease: A Prospective Cohort Study of 502,493 Participants From the UK Biobank. Biol Psychiatry. 1;93(9):790-801.

(3) Comment: Given the large number of tests, the multiple testing issue is concerning. To mitigate potential false positives, I recommend employing the Bonferroni threshold or FDR. The authors should only interpret exposures that are significant at the Bonferroni threshold.

We sincerely appreciate the reviewer's feedback. Thank you for pointing this out. We have added the results of the Bonferroni correction to the Statistical models for the causal effect inference section of the Methods section (page 10) in response to the reviewer's feedback.

The newly added text describing Bonferroni threshold is as follows:

“The above analyses were performed using the TwoSampleMR[41] package in the R (version 4.1.2) software. Association of exposures with outcomes was assessed using odds ratio (OR) and 95% confidence interval (95% CI), with OR > 1 indicating a positive association (risk factor) and 0 < OR < 1 indicating a negative association (protective factor). Differences with a two-sided p < .05 were considered statistically significant. Furthermore, owing to the relatively large number of exposure and outcome traits included in this study, the multiple testing correction method Bonferroni correction was added to identify significant hits, threshold for Bonferroni-corrected was 0.05 divided by 289,552 tests (p <1.727e-07).”

(4) Comment: In the discussion, the authors should interpret or highlight exposures that remain significant after multiple testing corrections.

Thank you for your valuable comment. In response to reviewer feedback, we have put extra emphasis on the exposures that remained significant after multiple testing corrections in the Discussion section (page 17). We thank you again for your feedback, as we believe that our manuscript has been improved substantially as a result of your input.

**Recommendations for the authors:**

**Reviewer #1 (Recommendations For The Authors):**
(1) Comment: In this study, the authors used the inverse-variance weighted (IVW) model as the major analysis method to perform Mendelian randomization analysis to identify various classes of risk or protective factors for AD, early-onset AD, and late-onset AD. An online database called MRAD has been thereby developed with the assistance of Shiny package. This study is a very intriguing study of great clinical and scientific significance that provided a thorough and comprehensive evaluation with regard to risk or protective factors for AD. It also provided physicians and scientists with a very convenient, free as well as user-friendly tool for further scientific investigation.I believe this manuscript is great research that is worth publishing with all the comments from the Public Review resolved.

We thank the reviewer for taking the time to read and provide valuable feedback on our manuscript, which allowed us to improve the overall quality of our research. All the comments from the Public Review have been rechecked, and appropriate changes have been made in accordance with the reviewers’ suggestions. Point-by-point responses to all the comments from the Public Review can be found in the above. If there are any further issues, please do not hesitate to let us know, so that we can ensure that our manuscript meets the high standards required for publication.

**Reviewer #2 (Recommendations For The Authors):**
(1) Comment: In the middle lower left section of the graphical abstract, the overlapping positive (N=63) and overlapping negative (N=16) do not sum to the overlapping number (N=80). Could you clarify if any have both positive and negative effects? Additionally, the font size inside the circular elements is too small to read.

We thank you for raising this issue. We have clarified this in the MRAD utility data mining section of Results section (page 12): A total of 63 exposure traits (risk factors) were positively associated with all the three main outcome traits, while 16 exposure traits (protective factors) were negatively associated with the three main outcome traits, with Ulcerative colitis (ebi-a-GCST000964) being negatively associated with the AD outcome traits of ieu-b-2 and ieu-a-297, and positively associated with the AD outcome traits of ieu-b-5067. Additionally, we apologize for the small, unreadable fonts in the graphical abstract figure. In response to reviewer feedback, we have increased the font size within the figure and enhanced the resolution to improve image readability (page 3).

(2) Comment: The x-axis label ("Alzheimer's disease outcome") should be more descriptive. If published GWAS results are used, indicate this as XXX et al. (2022). Also, specify the AD outcome for each category (e.g., AD, early-onset AD, late-onset AD). The y-axis labels should also be clarified; remove identification codes and retain only the exposure names. Apply the same improvements to Figures 2-8.

We appreciate your comments and suggestions for improving our manuscript.

(i) In response to reviewer feedback, information of published GWAS such as authors and year of publication have now been added to the x-axis labels, as demonstrated in Figure 4 (page 31).

(ii) The outcome IDs are unique. We used these IDs to represent the AD information on the x-axis to maintain a clean and clear figure. The corresponding details for each ID are explained in the Outcome traits section of the Methods section (page 8, as shown in Figure 2). AD_EO refers to early-onset AD, and AD_LO refers to late-onset AD, which are also specified in the Abbreviations (page 4).

(iii) We sincerely appreciate the reviewers’ meticulous feedback. While exposure IDs in this study are unique, exposure names are not. A single exposure name may correspond to multiple IDs, each with a potentially different source of information (e.g., author, year, population sample). We believe obtaining consistent results across multiple IDs further strengthens the reliability of our conclusions. Hence, for better clarity of specific exposure information, the exposure IDs have been retained.

(3) Comment: The results across Figures 1-8 are repetitive and not very informative. Consider other visualizations to condense the information into one or two figures. I would recommend using a Manhattan plot or PheWAS plot concept to effectively display many test results at once. Please display the Bonferroni threshold in the plot as a horizontal line to show which exposures are meaningful after adjusting multiple comparisons.

We appreciate this helpful suggestion. We have now condensed Figures 1–8 into a single figure (as shown in Figure 4). Additionally, we have now displayed the Bonferroni correction results in the sensitivity analysis results figures (as shown in Figure 5, Figure S1-S7).

(4) Comment: Consider placing Figure S1 as Figure 1, condensing Figures 1-8 into Figures 2 and 3, and placing the circular diagrams from Figure S6 as Figure 4.

We appreciate this valuable suggestion. The sequence of the figures has been adjusted.

(5) Comment: Create a main table summarizing robust and consistent exposures for AD that are significant at the Bonferroni threshold for readers. For each exposure, please include estimates from IVW, MR-Egger, weighted median, simple mode, weighted mode, maximum likelihood, and penalized weighted median, along with heterogeneity and horizontal pleiotropy tests. I would also highlight or bold estimates from the best-fit model/MR method to help readers identify the most reliable estimates when estimates from multiple methods are heterogeneous.

We appreciate this helpful suggestion. Owing to the excessive amount of information in the table, we have uploaded the table covering the aforementioned information according to the reviewer’s suggestion as supplementary materials (See Table S2). (i) The corresponding id.exposure that pass the Bonferroni threshold are reflected in red font. (ii) Furthermore, according to the customized best-fitted model (as mentioned in the Statistical models for causal effect inference section of Methods section), when there is no pleiotropy or when pleiotropy is not applicable (less than 3 SNPs), random-effects IVW model is the best model. These corresponding id.exposure are shown in red font with a yellow highlight. (iii) Moreover, according to the customized best-fitted model, when there is pleiotropy, MR-Egger is the best model. These corresponding id.exposure are shown in red font with a green highlight.

(6) Comment: Figures S4-S10: These figures are screenshots of web browsers and may not be worth showing. Consider using tools like Adobe AI or R ggplot to create more refined visualizations that are specific to the research question and improve the message of this work.

Thank you very much for your valuable suggestion in reviewing our manuscript. In this study, Figures S4-S10 are screenshots related to the user guide. We sincerely appreciate the reviewer’s feedback and have revised the subheading of this section to MRAD User Guide to clarify its purpose. Demonstrating both text and figures in this section, we aim to help users understand ways to operate MRAD more intuitively and easily.

(7) Comment: Additionally, please show upfront or highlight results from MR analyses based on R packages, as the author mentioned in the method section. Somehow it's difficult to find results from MR-Egger, weighted median, simple mode, weighted mode, maximum likelihood, and penalized weighted median, along with heterogeneity and horizontal pleiotropy tests in the supplementary materials. Apologies if I missed them. Please ensure these results are clearly presented.

We appreciate your comments and suggestions for improving our manuscript. Thank you for pointing this out. We have added the results of the sensitivity analysis based on R packages (as shown in Figure 5, Figure S1-S7, and Table S2).